# The Female-Biased General Odorant Binding Protein 2 of *Semiothisa cinerearia* Displays Binding Affinity for Biologically Active Host Plant Volatiles

**DOI:** 10.3390/biology13040274

**Published:** 2024-04-18

**Authors:** Jingjing Tu, Zehua Wang, Fan Yang, Han Liu, Guanghang Qiao, Aihuan Zhang, Shanning Wang

**Affiliations:** 1Key Laboratory of Environment Friendly Management on Fruit and Vegetable Pests in North China (Coconstructed by the Ministry and Province), Ministry of Agriculture and Rural Affairs, Institute of Plant Protection, Beijing Academy of Agriculture and Forestry Sciences, Beijing 100097, China; 15910506265@163.com (J.T.); wangzehua200707@163.com (Z.W.); evelynyangfan@163.com (F.Y.); 15230451215@163.com (H.L.); qghang98@126.com (G.Q.); 2College of Bioscience and Resources Environment, Beijing University of Agriculture, Beijing 102206, China; zhangaihuan@126.com

**Keywords:** *Semiothisa cinerearia*, general odorant binding protein, ligand-binding spectrum, molecular docking, electrophysiological, behavioral responses

## Abstract

**Simple Summary:**

The moth *Semiothisa cinerearia* (Lepidoptera: Geometridae) is a major pest of Chinese scholar trees (*Sophora japonica* L.). Olfaction is very important for insects to locate host plants and oviposition sites. Here, we identified the binding abilities of ScinGOBP2 for host plant volatiles using fluorescence-based competitive binding assays. We also confirmed the key amino acid residues that bind to plant volatiles in ScinGOBP2 via three-dimensional structure modeling and molecular docking. The ScinGOBP2 ligands had attractive or repellent behavioral effects on *S. cinerearia* for oviposition. Overall, ScinGOBP2 may play important roles in detecting host plant volatiles, and ScinGOBP2 ligands could be used as candidate olfactory regulators for the management of *S. cinerearia*.

**Abstract:**

Herbivorous insects rely on volatile chemical cues from host plants to locate food sources and oviposition sites. General odorant binding proteins (GOBPs) are believed to be involved in the detection of host plant volatiles. In the present study, one GOBP gene, *ScinGOBP2*, was cloned from the antennae of adult *Semiothisa cinerearia*. Reverse-transcription PCR and real-time quantitative PCR analysis revealed that the expression of *ScinGOBP2* was strongly biased towards the female antennae. Fluorescence-based competitive binding assays revealed that 8 of the 27 host plant volatiles, including geranyl acetone, decanal, *cis*-3-hexenyl n-valerate, *cis*-3-hexenyl butyrate, 1-nonene, dipentene, α-pinene and β-pinene, bound to ScinGOBP2 (*K*_D_ = 2.21–14.94 μM). The electrical activities of all eight ScinGOBP2 ligands were confirmed using electroantennography. Furthermore, oviposition preference experiments showed that eight host volatiles, such as decanal, *cis*-3-hexenyl n-valerate, *cis*-3-hexenyl butyrate, and α-pinene, had an attractive effect on female *S. cinerearia*, whereas geranyl acetone, 1-nonene, β-pinene, and dipentene inhibited oviposition in females. Consequently, it can be postulated that ScinGOBP2 may be implicated in the perception of host plant volatiles and that ScinGOBP2 ligands represent significant semiochemicals mediating the interactions between plants and *S. cinerearia*. This insight could facilitate the development of a chemical ecology-based approach for the management of *S. cinerearia*.

## 1. Introduction

Herbivorous insects primarily depend on their sense of smell to distinguish chemical signals released by host plants, enabling them to locate food sources and choose suitable places for laying eggs within complex chemical environments [1,2,3]. These chemical cues are presumably perceived by specialized olfactory sensilla, predominantly located on the insect antennae. The structure of olfactory sensilla is punctuated with numerous pores, forming a cavity filled with aqueous lymph. This cavity harbors olfactory receptor neurons’ (ORNs) dendritic branches and is enriched with small soluble proteins. Odorants present in the environment enter sensilla via cuticular pores, are dissolved in the sensillar lymph, initiate the activation of ORNs, and ultimately induce various behavioral responses [4,5,6]. Odorant binding proteins (OBPs) are the main types of soluble binding proteins involved in the binding, solubilizing, and transport of hydrophobic odorants across the aqueous environment of the sensillar lymph to odorant receptors in the dendritic membranes of ORNs [7,8,9].

In insects, OBPs are divided into different subfamilies based on their sequence and the tissue in which they are expressed [8,10,11]. Pheromone binding proteins (PBPs) and general odorant binding proteins (GOBPs) in Lepidoptera form a distinct subclass; they are abundant in moth antennae and are associated with chemosensory sensilla [12,13,14]. In Lepidoptera species, PBPs are believed to function in perception of female sex pheromones, while GOBPs are thought to be involved in interacting with plant odorants. For instance, GOBP2 of *Agrotis ipsilon* [15], GOBP2 of *Spodoptera frugiperda* [16], and GOBP1-2 of *Orthaga achatina* [17] can strongly bind to the corresponding host plant volatiles. Recently, genome editing via CRISPR–Cas9 was used to verify that the GOBPs of *S. litura* participate and *Conogethes punctiferalis* in the detection of host odorants in vivo [18,19]. These findings indicate that moth GOBPs play important roles in determining the location of the host plant. However, it has been reported in some studies that GOBPs can bind sex pheromones, and they are also thought to function in the perception of sex pheromones [15,20,21]; however, further in vivo functional studies are needed [18,22].

*Semiothisa cinerearia* (Bremer et Gray), which belongs to the subfamily Ennominae (Lepidoptera: Geometridae), is a forestry pest that attacks *Sophora japonica*. *S. japonica*, which is native to China and known as the Chinese scholar tree, widely cultivated in urban greenbelts in China [23,24]. In recent years, Chinese scholar trees have been severely damaged by *S. cinerearia* in northern China. To control *S. cinerearia*, excessive amounts of insecticides have been applied, which has had negative effects on the ecosystem. To devise environmentally friendly methods of pest control, such as by regulating olfactory behavior, many chemosensory genes, including 26 OBP genes [25,26] that await functional characterization, have been identified in *S. cinerearia*.

In the present study, we investigated the function of the female-biased GOBP, ScinGOBP2, in the detection of host plant volatiles by profiling the expression of *ScinGOBP2* transcripts in various tissues from male and female moths using reverse transcription PCR (RT–PCR) and quantitative real-time PCR (qRT–PCR). ScinGOBP2 was expressed in vitro to evaluate its binding affinity for 27 volatiles from *S. japonica* volatiles via fluorescence binding assays. Additionally, homology modeling and molecular docking were utilized to identify the essential amino acids in ScinGOBP2 involved in ligand binding. The biological activity of the ScinGOBP2 ligands was further validated through electroantennography (EAG) and two-choice oviposition preference assays.

## 2. Materials and Methods

### 2.1. Insect Culture, Tissue Collection, RNA Extraction, and cDNA Synthesis

Pupae of *S. cinerearia* were harvested from soil beneath Chinese scholar trees in the suburbs of Beijing, China. The pupae were maintained in a rearing cage at 25 ± 1 °C and 60 ± 10% relative humidity with a 16L:8D photoperiod. The emerged adults were fed a 10% sucrose solution. For *ScinGOBP2* gene cloning and tissue expression profiling, male antennae, female antennae, heads (without antennae), legs and bodies (a mixture of thoraxes, abdomens, and wings) were excised from 1- to 3-day-old adult moths. All samples were immediately frozen in liquid nitrogen and subsequently stored at −80 °C for RNA extraction. Total RNA was isolated from the samples using TRIzol reagent (Invitrogen, Carlsbad, CA, USA), adhering to the protocol provided by the manufacturer. The integrity and quantity of the extracted RNA samples were verified through 1.2% agarose gel electrophoresis and measurement with a NanoPhotometer N60 (Implen, München, Germany), respectively. cDNA was generated from 2 μg of RNA for each sample using the Fast Quant RT Kit (with gDNase) (Tiangen, Beijing, China), according to the manufacturer’s instructions, and this cDNA served as the template for subsequent gene cloning and RT–PCR and qRT–PCR analyses.

### 2.2. Gene Cloning and Sequence Analysis

Drawing on the adult *S. cinerearia* antennal transcriptome data [25], the open reading frame (ORF) of *ScinGOBP2* was cloned using gene specific primers (Appendix A). PCR was performed in 25 μL reactions containing 2 × Premix TaqTM (12.5 μL; Biomed, Beijing, China), forward primers (10 μM), reverse primers (10 μM), the antennal cDNA template (1 μL), and ddH_2_O (9.5 μL). PCR was conducted as follows: 94 °C for 4 min, followed by 30 cycles of 94 °C for 30 s, annealing at 55 °C for 30 s, and extension at 72 °C for 45 s. The final extension step was at 72 °C for 5 min. The PCR products were checked by using 1.2% agarose gel and subsequently verified via DNA sequencing.

The signal peptides were predicted using the SignalP5.0 server (http://www.cbs.dtu.dk/services/SignalP/) (accessed on 24 June 2022). The isoelectric points (IPs) and molecular weights (MWs) were calculated using the ExPASy Proteomics Server (https://www.ExPASy.org/) (accessed on 24 June 2022). Sequences were aligned using ClustalW Multiple Alignment in BioEdit version 7.1.3.0.

### 2.3. RT–PCR and qRT–PCR

The primers used for RT–PCR and qRT–PCR analyses were designed with Primer 3 (http://primer3.ut.ee/) (accessed on 26 November 2022) (Appendix A).

The differential expression of *ScinGOBP2* across various tissues in both male and female moths was examined using RT–PCR, employing Taq DNA polymerase (Biomed, Beijing, China) for the reactions. For each 25 μL reaction, 200 ng of cDNA from different tissues served as a template. The PCR was conducted with the same cycling conditions outlined above. *β-Actin* from *S. cinerearia* was used as the control gene to test the integrity of the cDNA. The PCR products were checked by using 1.2% agarose gels, and a randomly chosen PCR product was sequenced to confirm its identity.

The relative transcript abundance of *ScinGOBP2* in different tissues was determined via qRT–PCR on an ABI Prism 7500 Fast Detection System (Applied Biosystems, Carlsbad, CA, USA). Two endogenous genes, *β-actin* and *TBP* (TATA-binding protein-associated factor 172), were used for normalization. The efficiency of the primers was calculated by analyzing the standard curves with a 5-fold dilution series of the female antennal cDNA template. The 25 μL reactions contained 2 × SuperReal PreMix Plus (12.5 μL; TianGen, Beijing, China), 50 × ROX Reference Dye (0.5 μL), forward primers (7.5 μM), reverse primers (7.5 μM), sample cDNA (200 ng), and ddH_2_O (8.5 μL). The qRT–PCR cycling parameters consisted of 95 °C for 15 min, followed by 40 cycles of 95 °C for 10 s and 60 °C for 32 s. Melting curves were constructed by raising the temperature to 95 °C in 0.35 °C/s increments. Non-template controls were included in each experiment. Each qRT–PCR experiment was carried out in three biological replicates and three technical replicates. The comparative cycle threshold (2^−ΔΔCT^) method was used for measuring the relative transcript levels in each tissue.

### 2.4. Recombinant ScinGOBP2 Expression

The chemically synthesized cDNA was cloned and inserted into pET30a (+) vector by Taihe (Beijing, China). The expression plasmid was transformed into Rosetta (DE3) competent cells. The protein was expressed in LB broth at 18 °C for 16 h through induction with 1 mM isopropyl-β-D-thiogalactopyranoside. The bacterial cells were harvested via centrifugation and resuspended in PBS (pH 7.4, 10 mM). After sonication and centrifugation, the recombinant protein was detected in both the supernatant and inclusion bodies. The supernatant was applied to the Ni column (GenScript, Nanjing, China) to purify the protein. The His-tag was removed using recombinant enterokinase (Yaxin, Shanghai, China), adhering to the manufacturer’s guidelines. The purified ScinGOBP2 was dialyzed in PBS, and its concentration was then measured using the Bradford assay.

### 2.5. Fluorescence-Based Competitive Binding Assays

A total of 27 volatile compounds were selected for a fluorescence-based competitive binding assay based on previously reported information for *S. cinerearia* host plants [27]. This experiment was performed on a spectrofluorometer F-380 (Tianjin, China) using slits of 10 nm and a light path of 1 cm. The fluorescent probe N-phenyl-1-naphthylamine (1-NPN) was excited at 337 nm, and emissions were recorded between 390 and 530 nm. The affinity of 1-NPN for ScinGOBP2 was measured by titrating a 2 μM solution of the protein with aliquots of 1 mM 1-NPN in methanol to final concentrations of 1–16 μM. The dissociation constants of 1-NPN and the proteins were calculated using the software GraphPad Prism 8.0. Competitive binding was measured via the titration of the protein/1-NPN (both at 2 μM) mixture by adding aliquots of 1 mM of methanol solution of ligand to final concentrations of 2–30 μM. Dissociation constants of the competitors were calculated using the equation *K*_D_ = IC_50_/(1 + [1-NPN]/*K*_1-NPN_), where IC_50_ represents the ligand concentration achieving a 50% decrease in 1-NPN’s initial fluorescence intensity, [1-NPN] represents the free concentration of 1-NPN, and *K*_1-NPN_ represents the dissociation constant of the complex ScinGOBP2/1-NPN. These procedures were replicated three times, except for the ligand interactions that showed no significant binding, which were assessed in singular assays.

### 2.6. Three-Dimensional Modeling of ScinGOBP2 and Ligand Docking

A three-dimensional (3D) structure of ScinOBP2 was constructed using AlphaFold2 by submitting the amino acid sequence to the Beijing Super Cloud Computing Center (Beijing, China). The 3D structure was then subjected to molecular dynamics (MD) simulation using the Amber22 and AMBER ff19SB force fields for energy minimization, resulting in the refinement of its structure. The qualities of the optimized models were evaluated using the ERRAT and PROCHECK programs. The ligands were subjected to 3D optimization in ChemDraw 3D (23.0) and refined via energy minimization. Molecular docking was performed using AutoDock Vina (4.2.6). The top models were selected according to the lowest free binding energy (kcal/mol) and visually analyzed using PyMOL (2.5.5).

### 2.7. Electrophysiological Recordings

Electroantennogram recordings were carried out to measure the antennal responses of *S. cinerearia* to ScinGOBP2 ligands. The antennae of *S. cinerearia* adults were carefully removed from the base, and the tips were carefully cut off. The prepared antennae were affixed to electrode holders using conductive electrode gel. Subsequently, a 10 μL aliquot of the test compound (10 μg/μL, diluted in paraffin oil) was applied to a strip of filter paper. This strip was then placed into a 1000 µL pipette tip, serving as a cartridge. The test cartridge was connected to a stimulus controller (CS-55; Syntech, Kirchzarten, Germany) that delivered a 0.5 s stimulus every 30 s at a constant flow rate of 10 mL/s. Antennal signals were recorded using an EAG Pro system (Syntech). In all the experiments, antennae were exposed to a solvent control (paraffin oil) at the beginning and end of a series of sample measurements. Test stimuli were presented in a randomized sequence, interspersed between two control puffs. The EAG responses to the test odors were adjusted by subtracting the mean amplitude of the two control signals. EAG responses were measured from a cohort of twelve insects, each with different antennae (*n* = 12).

### 2.8. Two-Choice Oviposition Preference Assay

The oviposition preference of female *S. cinerearia* for the ScinGOBP2 ligands was assessed in a screened enclosure (length/width/height = 86 cm:46 cm:48 cm). To ensure mating of all the female moths, newly emerged males and females were cohabitated in a rearing enclosure (30 cm × 30 cm × 30 cm) at a sex ratio of 1:2 for a period of 2 days. Mated females for subsequent experiments were randomly selected from this enclosure. Each screened cage housed five gravid females. For adult nourishment, a small Petri dish containing 10% honey solution was positioned at the cage’s base center. Test compounds (100 μg, diluted in paraffin oil) or the solvent control (paraffin oil) were applied to a cotton ball, which was then affixed to the opposite inner sides of the cage. After 48 h, the number of eggs deposited on the gauze on both the treatment and control sides was counted. The oviposition preference index was determined using the formula (T − C)/(T + C), where T represents the egg count on the treatment side and C represents the count on the control side. Each experiment was replicated eight times.

### 2.9. Statistical Analysis

The statistical analysis of the data was conducted using SPSS 18.0 and GraphPad Prism 8.0. Data are expressed as mean ± standard error (SE). Statistically evaluation was performed by utilizing one-way analysis of variance (ANOVA), followed by Tukey’s honestly significant difference (HSD) test (significance level: *p* < 0.05).

## 3. Results

### 3.1. Gene Cloning and Sequence Analysis of ScinGOBP2

The nucleotide sequence of ScinGOBP2 was confirmed via molecular cloning and sequencing. The ORF of ScinGOBP2 spans 519 nucleotides, encoding a polypeptide of 172 amino acid residues, consistent with earlier reports. The N-terminus of ScinGOBP2 is anticipated to harbor a signal peptide composed of 31 amino acid residues. The mature ScinGOBP2 protein is predicted to have molecular weight of 16.00 kDa and an isoelectric point of 5.16. ScinGOBP2 has a typical six-cysteine signature and exhibits the motif pattern C_1_-X_15-39_-C_2_-X_3_-C_3_-X_21-44_-C_4_-X_7-12_-C_5_-X_8_-C_6_ of typical insect OBPs. Multiple sequence alignment illustrated that the ScinGOBP2 protein presented high homology across several lepidopterans, with the highest similarity to *Cydia pomonella* GOBP2 (CpomGOBP2, 84.40%) (Figure 1).

### 3.2. Tissue Expression Patterns of ScinGOBP2

RT–PCR was utilized to assess the tissue-specific expression patterns of ScinGOBP2 transcripts across various adult tissues. The presence of β-actin in all tissues served as confirmation the quality of the cDNA templates in each sample. ScinGOBP2 expression was predominantly observed in the antennae of both sexes, with a reduced intensity of PCR bands observed in male antennae compared to female antennae (Figure 2). The expression level of the ScinGOBP2 transcript was significantly higher in the antennae compared to other tissues. Specifically, the ScinGOBP2 transcript was expressed at approximately 18,545 and 8263 times higher in the female and male antennae, respectively, than in each of other body parts. Furthermore, the expression levels in female antennae were approximately 2.2-fold higher than in male antennae (Figure 2).

### 3.3. Binding Characteristics of Recombinant ScinGOBP2

To identify potential ligands for ScinGOBP2, we initially expressed and purified mature ScinGOBP2 without any modifications. The size and purity of the resultant recombinant protein were verified through SDS–PAGE (Appendix A). Competitive binding assays were performed to investigate its affinity for 27 volatile compounds, using 1-NPN as a fluorescent probe. ScinGOBP2 binds 1-NPN with good affinity (Figure 3A,B), and the dissociation constant (*K*_D_) for the ScinGOBP2/1-NPN complexes was determined to be 2.27 μM. ScinGOBP2 exhibited strong bounding to geranyl acetone, decanal, *cis*-3-hexenyl n-valerate, *cis*-3-hexenyl butyrate, 1-nonene, dipentene, α-pinene, and β-pinene, with *K*_D_ values ranging between 2.21 μM and 14.94 μM (Figure 3C, Table 1).

### 3.4. Protein Structure Prediction and Molecular Docking

The 3D structure of ScinGOBP2 (Appendix A) was generated using Alphafold2. The protein model underwent a 20 ns MD simulation for energy minimization and protein stabilization. The structural stability of the protein was assessed by calculating the root mean square deviation (RMSD) and root mean square fluctuation (RMSF) (Appendix A). ERRAT and PROCHECK were used to evaluate the qualities of the protein model. The ERRAT results revealed an overall quality factor of 97.85 (Appendix A). Moreover, the Ramachandran plot indicated that 89.2% of the residues were in the most favored region (Appendix A). All the parameters indicated that the 3D modeling of ScinGOBP2 was reasonable and reliable.

The docking analysis revealed that the ligands demonstrated significant binding affinity for ScinGOBP2, with binding energy values ranging from −5.3 to −7.3 kcal/mol (Table 2). Geranyl acetone had the highest interaction energy, marked by a binding energy of −7.3 kcal/mol (Table 2). Following this, dipentene showed notable interaction, with a binding energy of −6.8 kcal/mol, involving π–sigma interactions with the Phe31 residue (Figure 4, Table 2). The Ser75 amino acid residue in ScinGOBP2 was involved in forming hydrogen bonds with decanal, *cis*-3-hexenyl n-valerate, and *cis*-3-hexenyl butyrate (Figure 4). Hydrophobic interactions were observed between all compounds and ScinGOBP2 (Figure 4, Table 2).

### 3.5. EAG Recordings

To assess the biological activity of ScinGOBP2 ligands, electrophysiological responses of both female and male *S. cinerearia* to the ScinGOBP2 ligands were evaluated using EAG recordings. All eight volatile compounds elicited EAG responses in the antennae of both sexes of *S. cinerearia* (Figure 5). Females exhibited the most robust EAG response to dipentene, whereas males showed the strongest EAG response to decanal, with mean response values of 0.086 and 0.063 mV, respectively. The EAG responses to all the volatiles, except for dipentene, did not show significant differences between adult males and females.

### 3.6. Two-Choice Oviposition Assays

To study the influence of ScinGOBP2 ligands on *S. cinerearia* oviposition, two-choice oviposition assays were conducted in a screened cage (Figure 6A). Females demonstrated a significant preference for laying eggs on gauze areas exposed to decanal, *cis*-3-hexenyl n-valerate, *cis*-3-hexenyl butyrate, and α-pinene compared to other odors (Figure 6B). On the other hand, geranyl acetone, 1-nonene, dipentene, and β-pinene were found to deter *S. cinerearia* from ovipositing (Figure 6B).

## 4. Discussion

Moths have a well-developed olfactory system that enables them to utilize different chemical cues for mating and locating a host. Moth PBPs/GOBPs were the first OBPs studied and have widely documented associations with female sex pheromones and host plant volatile detection. They form a Lepidoptera-specific subfamily within insect OBP gene family, although some non-Lepidoptera OBPs are functionally similar to GOBPs/PBPs and named PBPs or GOBPs [13,28]. In this study, the *ScinGOBP2* gene was isolated from *S. cinerearia*. ScinGOBP2 exhibits six conserved cysteines characteristics of the OBP family and shares a high sequence identity with GOBP2s from other insect species [8,11]. The expression patterns of moth GOBPs, showing sexual dimorphism, vary among different species. For *S. cinerearia*, ScinGOBP2 was predominantly expressed in the antennae, with significant higher expression levels in females compared to males. This suggests that ScinGOBP2 might be involved in female-specific behaviors. Female-biased expression has also been observed in several other moth species, such as *Lobesia botrana* [29], *Conogethes pinicolalis* [30], and *Histia rhodope* [31].

The results from the fluorescent binding assays showed that ScinGOBP2 specifically bound to the tested host plant volatiles, with a high binding affinity (*K*_D_ below 15 μM) for geranyl acetone, decanal, *cis*-3-hexenyl n-valerate, *cis*-3-hexenyl butyrate, 1-nonene, dipentene, α-pinene, and β-pinene. Similar results for GOBP2 binding were found in other moths. AipsGOBP2 exhibited a high binding affinity for the plant volatiles *cis*-3-hexen-1-ol, oleic acid, dibutyl phthalate, and β-caryophyllene [15], and OachGOBP2 had a high binding affinity for the host plant volatiles farnesol and α-phellandrene [17]. Prior research has indicated that certain specific amino acids situated within hydrophobic cavities may play a role in the ligand-binding processes of insect OBPs. For example, in BminOBP3, V120 is involved in undecanol binding [32], while in LstiGOBP1, Thr15 and Trp43 are involved in binding plant volatiles [33]. Molecular docking studies have shown that ScinGOBP2 strongly interacts with its ligands. Ser75 forms hydrogen bonds with decanal, *cis*-3-hexenyl n-valerate, and *cis*-3-hexenyl butyrate, indicating that it can actively participate in binding in ScinGOBP2. However, additional research is required to pinpoint the precise binding sites that mediate the interactions between ScinGOBP2 and its ligands. Therefore, future experiments involving site-directed mutagenesis will be crucial for addressing this question.

The ScinGOBP2 ligands can elicit antennal responses in both male and female individuals, indicating that these volatile compounds released by plants might serve as potential semiochemicals for *S. cinerearia*. Plant volatiles such as dipentene, decanal, *cis*-3-hexenyl n-valerate, and geranyl acetone have been shown to act as repellents and/or attractants for some insects [34,35,36,37,38]. In the present study, different ScinGOBP2 ligands at the same concentrations elicited different behaviors in *S. cinerearia*. Among these volatiles, four (decanal, *cis*-3-hexenyl n-valerate, *cis*-3-hexenyl butyrate, and α-pinene) elicited significant attractant behavioral responses, and four (geranyl acetone, 1-nonene, β-pinene, and dipentene) elicited marked repellent behavioral responses, suggesting that these plant volatiles are vital semiochemicals in communication between female *S. cinerearia* individuals and their host plants in natural habitats. In nature, some male moths may also use plant volatiles to find mates [39,40,41]. Although ScinGOBP2 ligands can evoke EAG responses in the antennae of male *S. cinerearia*, there was no apparent correlation between the antennal response to compounds in the EAG analysis and the compounds’ behavioral effects [42,43,44]. Therefore, further studies are needed to determine whether ScinGOBP2 ligands have behavioral effects on male *S. cinerearia*.

## 5. Conclusions

We document the female-biased expression and ligand-binding ability of ScinGOBP2 from *S. cinerearia*, offering insights into the potential olfactory function of GOBP2 in the host-seeking behavior of *S. cinerearia*. Our behavioral trials showed that decanal, *cis*-3-hexenyl n-valerate, *cis*-3-hexenyl butyrate, and α-pinene may represent novel attractants for *S. cinerearia*. Further research is needed to validate the behavioral effects of these attractants in the field and explore their potential applicability in monitoring and managing *S. cinerearia* populations.

## Figures and Tables

**Figure 1 biology-13-00274-f001:**
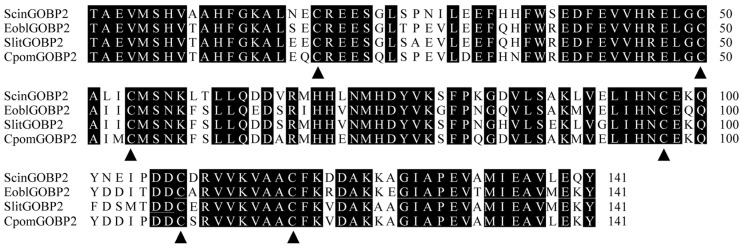
Alignment of amino acid sequences of ScinGOBP2. Only the mature proteins were aligned. The black triangles show the six highly conserved cysteine residues. EoblGOBP2 (*Ectropis obliqua*, ACN29681.1); SlitGOBP2 (*S. litura*, XP_022817877.1); CpomGOBP2 (*C. pomonella*, AFP66958.1).

**Figure 2 biology-13-00274-f002:**
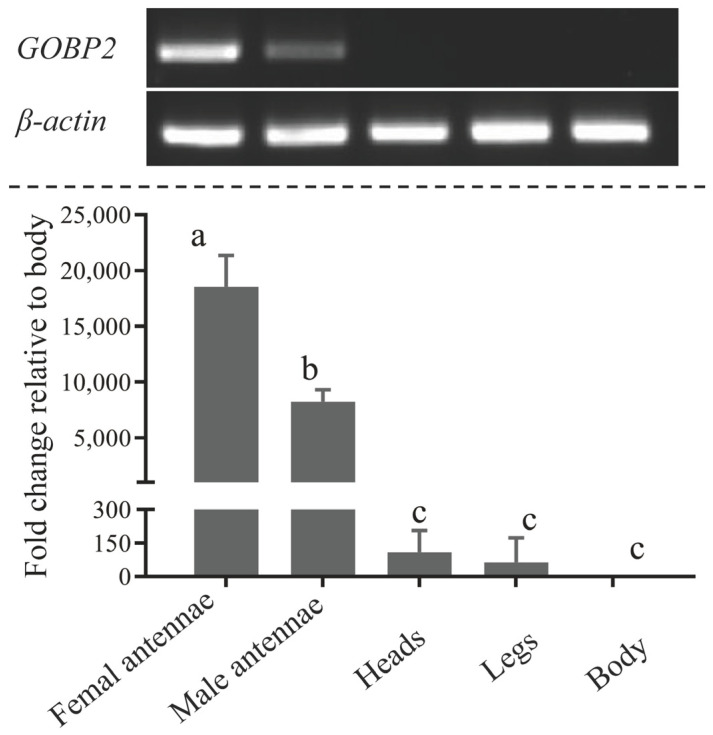
*ScinGOBP2* transcript levels in different tissues were assessed via RT–PCR and qRT–PCR. The error bars represent the standard error, and the different letters indicate significant differences (*p* < 0.05).

**Figure 3 biology-13-00274-f003:**
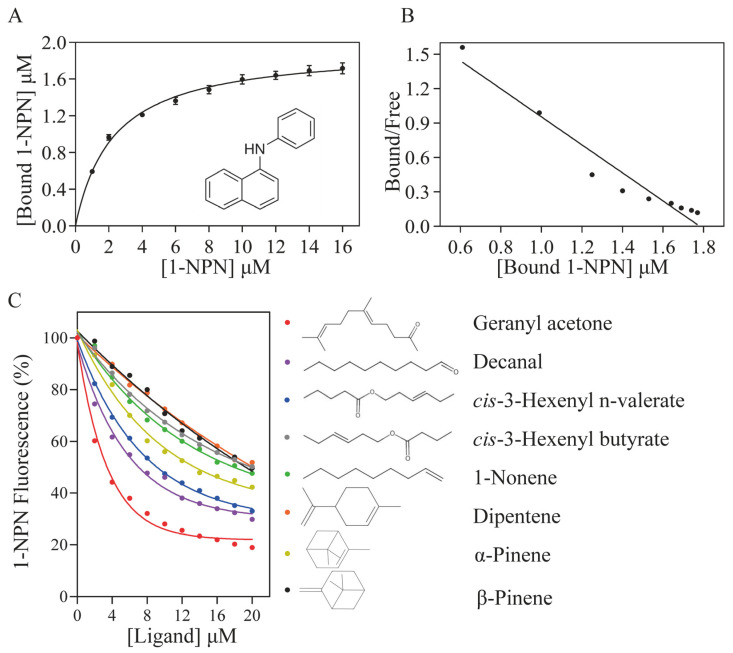
Characteristics of binding interaction exhibited by ScinGOBP2. The affinity of ScinGOBP2 to the fluorescent probe 1-NPN is illustrated through a binding curve (**A**) and Scatchard plot (**B**). (**C**) depicts competitive binding curves that demonstrate the binding of ScinGOBP2 to the selected ligands.

**Figure 4 biology-13-00274-f004:**
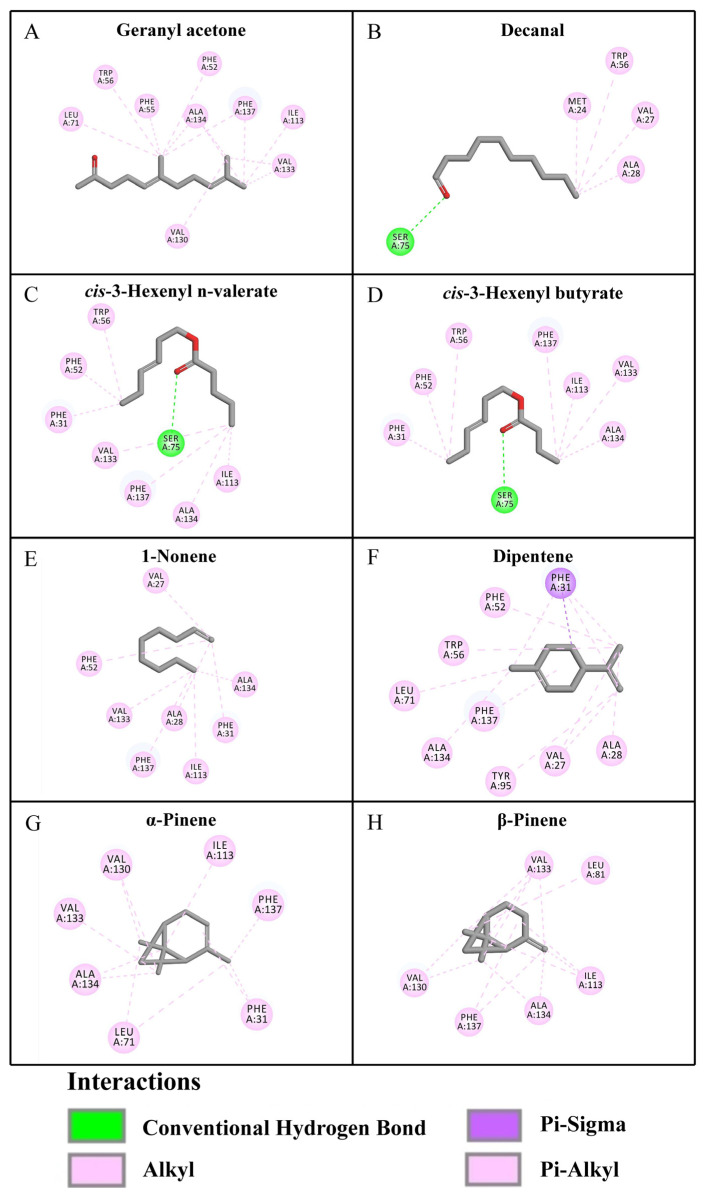
Molecular docking of ScinGOBP2 with (**A**) geranyl acetone, (**B**) decanal, (**C**) *cis*-3-hexenyl n-valerate, (**D**) *cis*-3-hexenyl butyrate, (**E**) 1-nonene, (**F**) dipentene, (**G**) α-pinene, and (**H**) β-pinene.

**Figure 5 biology-13-00274-f005:**
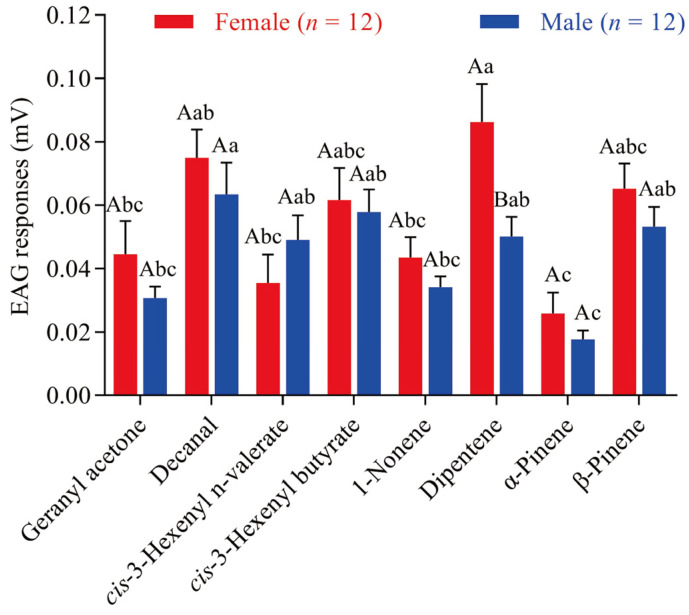
EAG responses of female and male *S. cinerearia* antennae in response to various ligands of ScinGOBP2. Distinct uppercase letters denote significant differences between females and males, while distinct lowercase letters signify significant differences among different chemicals (*p* < 0.05).

**Figure 6 biology-13-00274-f006:**
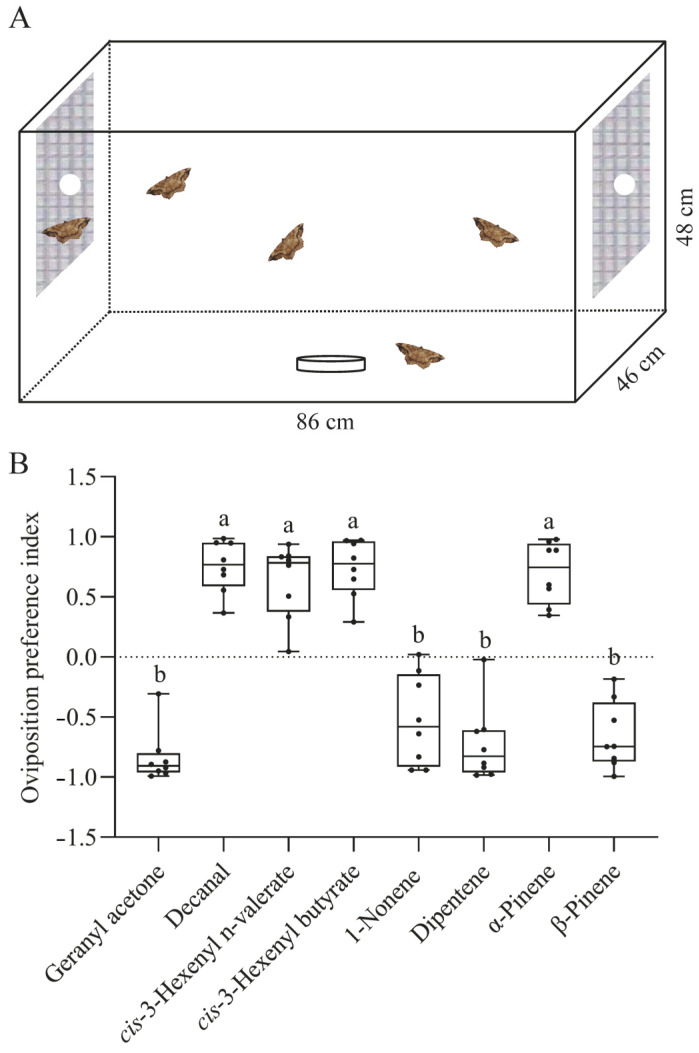
Oviposition preference of female *S. cinerearia* adults for ScinGOBP2 ligands. (**A**) Schematic diagram of the two-choice oviposition assay. (**B**) Oviposition preference index of females towards eight compounds. Different letters represent significant differences (*p* < 0.05).

**Table 1 biology-13-00274-t001:** Binding affinities of all the tested ligands for ScinGOBP2.

Ligand	Source	CAS Number	Purity (%)	*K*_D_ (μM) *
Geranyl acetone	Macklin	3796-70-1	≥98%	2.21 ± 0.17
Acetophenone	Macklin	98-86-2	≥99.5%	–
*cis*-3-Hexen-1-ol	Macklin	928-96-1	98%	–
*cis*-2-Nonen-1-ol	TCI	41453-56-9	>95.0%	–
2-Ethylhexan-1-ol	TCI	104-76-7	>99.5%	–
1-Octanol	TCI	111-87-5	>99%	–
Hexanal	TCI	66-25-1	>98%	–
Nonanal	TCI	124-19-6	>95%	–
Decanal	TCI	112-31-2	>97%	5.07 ± 0.27
3-Methyl-2-butenal	Macklin	107-86-8	98%	–
2-Hexenal	Macklin	6728-26-3	98%	–
Octane	Macklin	111-65-9	>99%	–
*cis*-3-Hexenyl acetate	Macklin	3681-71-8	98%	–
*cis*-3-Hexenyl n-valerate	Macklin	35852-46-1	98%	6.55 ± 0.18
Ethyl acetate	Macklin	141-78-6	≥99.7%	–
Butyl acetate	Macklin	123-86-4	≥99.7%	–
Hexyl acetate	TCI	142-92-7	>99%	–
*cis*-3-Hexenyl butyrate	Macklin	16491-36-4	≥98%	14.17 ± 0.04
Nonanoic acid	Macklin	112-05-0	>99%	–
Capric Acid	Macklin	334-48-5	>99%	–
2-octeno	TCI	111-67-1	>95%	–
1-Nonene	Macklin	124-11-8	95%	12.20 ± 0.44
Dipentene	Macklin	7705-14-8	95%	14.94 ± 0.77
Ocimene	Macklin	13877-91-3	≥90%	–
α-Pinene	Macklin	80-56-8	98%	9.41 ± 0.28
β-Pinene	Macklin	127-91-3	≥95%	13.19 ± 0.19
Isoprene	Macklin	78-79-5	>99%	–

* Dissociation constant (*K*_D_) values are reported only where IC_50_ values could be measured. “–” indicates that data are not available.

**Table 2 biology-13-00274-t002:** Docking results for ScinGOBP2 with different ligands.

Ligand	Binding Energy (kcal/mol)	Hydrophobic Interactions
Geranyl acetone	−7.3	Phe52, Phe55, Trp56, Leu71, Ile113, Val130, Val133, Ala134, Phe137
Decanal	−5.7	Trp56, Met24, Val27, Ala28
*cis*-3-Hexenyl n-valerate	−6.2	Phe31, Phe52, Trp56, Ile113, Val133, Ala134, Phe137
*cis*-3-Hexenyl butyrate	−5.9	Phe31, Phe52, Trp56, Ile113, Val133, Ala134, Phe137
1-Nonene	−5.3	Val27, Phe52, Val133, Phe137, Ala28, Ile113, Phe31, Ala134
Dipentene	−6.8	Phe52, Trp56, Leu71, Phe137, Ala134, Tyr95, Val27, Ala28
α-Pinene	−6.2	Phe31, Leu71, Ile113, Val130, Val133, Ala134, Phe137
β-Pinene	−6.3	Phe31, Leu71, Ile113, Val130, Val133, Ala134, Phe137

## Data Availability

The original contributions presented in the study are included in the article and Appendix A; further inquiries can be directed to the corresponding author.

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
