# Peer review of "The Female-Biased General Odorant Binding Protein 2 of *Semiothisa cinerearia* Displays Binding Affinity for Biologically Active Host Plant Volatiles"

_biology, 2024, doi:10.3390/biology13040274_

Round 1

Reviewer 1 Report

Comments and Suggestions for Authors

   The reviewer has read with interest the manuscript entitled as “The female-biased general odorant binding protein 2 of Semiothisa cinerearia displays binding affinity for biologically active host plant volatiles” submitted by Jingjing Tu, Zehua Wang, Fan Yang, Han Liu, Guanghang Qiao, Aihuan Zhang, Shanning Wang to the science journal “Biology”. GOBPs which are mainly found in Lepidoptera are crucial in olfaction and have been examined mainly with molecular biological methods. The authors cloned with standard techniques ScinGOBP2 from the antennae of Semiothisa cinerearia, revealed with a fluorescent probe that eight of the 27 host plant volatiles bound to ScinGOBP2 and identified key amino acid residues that bind to plant volatiles in ScinGOBP2 via three-dimensional structure modelling and molecular docking. Further, they found with behavioral and electrophysiological experiments that four of the eight volatiles of host plants were attractive and the remainders were inhibitory and concluded that ScinGOBP2 might be involved in the detection of host plant volatiles. The experiments were conducted with the standard methods and the obtained data were treated almost adequately using standard network applications.

While the manuscript is well written, the reviewer hopes that the authors will discuss what is the role of GOBPs in olfactory behavior of insects and discuss why GOBPs have mainly developed in lepidoptera, if possible.

As some minor problems are included in the manuscript as follows, treat them adequately.

Line 28       The electcal -> The electrical

Line 51       are the main type of soluble binding -> are the main types of soluble binding

Line 64       in some research -> in some researches

Line 190-191     In all the experiments, the antennae were stimulated with solvent control at the beginning and the end of all the sample measurements. ->(question) Does solvent control mean paraffin-oil without any odorants?

Line 240     in the female and male antennae than in other body parts -> in the female and male antennae than in each of other body parts

Figure 6      The letters showing the significant differences seem to be inadequate. Although the response to geranyl acetone appears very different from the response to decanal, the letters above the bars are the same “b”. Is it true?

Line 313-314     GOBPs, a subfamily of lepidopteran-specific OBPs -> (Question) Are GOBPs lepidopteran-specific? See E. Lescop et al Biochemistry 2009, 48, 2431–2441. They showed that a species of honeybee has GOBP.

Comments on the Quality of English Language

Almost good.

Reviewer 2 Report

Comments and Suggestions for Authors

This is a well written article chararacterising the gOBP2 of Semiothisa cinerearia. The fluorescent binding experiments with NPN need to be re-examined -Figure 3. The first point measured was at 2uM but it looks like the Kd is lower than that and more points are needed at the lower concentrations.  The Scatchard plot may not be the best method of determining the affinity constant and the fit is poor due to the bias induced by the first point. There are a few minor typographical errors - eg. electrical in the abstract. The article represents a good body of work and the authors must be congratulated. 

Comments on the Quality of English Language

Very minor typographical errors.
